# Design and Preliminary Experiment of W-Band Broadband TE$_{02}$ Mode Gyro-TWT

**Xu Zeng** [1,*], **Chaohai Du** [2,*] , **An Li** [1], **Shang Gao** [1], **Zheyuan Wang** [1], **Yichi Zhang** [1], **Zhangxiong Zi** [1] **and Jinjun Feng** [1]

1   National Key Laboratory of Science and Technology on Vacuum Electronics, Beijing Vacuum Electronics Research Institute, Beijing 100015, China; youhunhaha123@163.com (A.L.); gaoshang19941210@163.com (S.G.); 18200123346@163.com (Z.W.); zyc913374137@163.com (Y.Z.); Zhangxiongzi@163.com (Z.Z.); fengjinjun@tsinghua.org.cn (J.F.)
2   School of Electronics Engineering and Computer Science, Peking University, Beijing 100871, China
*   Correspondence: zengxu1108@163.com (X.Z.); duchaohai@pku.edu.cn (C.D.)

**Abstract:** The gyrotron travelling wave tube (gyro-TWT) is an ideal high-power, broadband vacuum electron amplifier in millimeter and sub-millimeter wave bands. It can be applied as the source of the imaging radar to improve the resolution and operating range. To satisfy the requirements of the W-band high-resolution imaging radar, the design and the experimentation of the W-band broadband TE$_{02}$ mode gyro-TWT were carried out. In this paper, the designs of the key components of the vacuum tube are introduced, including the interaction area, electron optical system, and transmission system. The experimental results show that when the duty ratio is 1%, the output power is above 60 kW with a bandwidth of 8 GHz, and the saturated gain is above 32 dB. In addition, parasitic mode oscillations were observed in the experiment, which limited the increase in duty ratio and caused the measured gains to be much lower than the simulation results. For this phenomenon, the reasons and the suppression methods are under study.

**Keywords:** broadband; gyro-TWT; high-resolution imaging radar; TE$_{02}$ mode; W-band

## 1. Introduction

The gyro-TWT is a vacuum electron amplifier based on the principle of the relativistic electron cyclotron maser. It can be used as the source to generate the high-power (kW levels) and broadband RF output in millimeter and sub-millimeter wave bands [1–5]. It is suitable as an important component of the transmitter, which can be applied to the radar, telecommunication, etc. Especially in W-band, the high-power, high-resolution imaging radar plays an important role in the applications of satellite imaging and deep space detection. A Haystack Ultra-wideband Satellite Imaging Radar (HUSIR) has been developed in the United States, in which a W-band gyro-TWT has been used as a part of the amplification link [6,7]. In order to achieve the detectability of an object with a diameter of 10 cm, the required operating bandwidth and minimum output power of the radar are 8 GHz and 100 kW, respectively. In the HUSIR, a W-band gyro-TWT with a bandwidth of 8 GHz and output power of 1 kW has been used to drive 16 gyrotwystrons to achieve the required capabilities. Therefore, under the precondition of sufficient bandwidth, increasing the output power is beneficial to the practical application of the gyro-TWT.

Reviewing the previous experiment results of the W-band gyro-TWT [8–13], all the designs adopt the fundamental harmonic of the TE$_{01}$ cylindrical waveguide mode as the operating mode, because of its high beam–wave interaction efficiency. At CPI (Communication & Power Industries), the designed tube has achieved an output power of 1.5 kW with a bandwidth of 8 GHz [8,9], which has been applied to the HUSIR. At UC Davis, the designed tube based on the heavily loaded and short copper stage interaction region achieved an output power of 140 kW, but the bandwidth was only 2 GHz [9–13]. At

IAP (Institute of Applied Physics, Russian Academy of Sciences), a W-band high-gain gyro-TWT based on the helical-waveguide was presented. The greatest advantage of this is that it greatly reduces the requirement of operating magnetic field intensity. The PIC simulations predict the maximum output power of 3 kW at about 95 GHz and a 1-kW-level bandwidth of 7.2 GHz when driving by a 100 mW input signal [14–18]. In China, research on W-band gyro-TWTs has been receiving more and more attention in recent years. At UESTC (University of Electronic Science and Technology of China), the periodic lossy circuit has been applied to the interaction region with an output power of 112 kW and a bandwidth of 3.8 GHz being achieved in the experiment [19,20]. At BVERI (Beijing Vacuum Electronics Research Institute), we also used the periodic lossy circuit as the interaction region of the tube, and obtained an output power of 110 kW and a bandwidth of 4 GHz [21].

However, an inner groove on the tube wall caused by the bombardment of the electrons had been observed in a previous experiment. The phenomenon indicates that a poor electron beam transmission seriously limits the output power of the tube. According to the results of theoretical calculations, the interaction cavity radius of the $TE_{01}$ mode is only 2 mm, and the distance between the electron beam and the tube wall is less than 1 mm. It is very hard to avoid the bombardment of the electrons. Consequently, improvement of the electron beam transmission and the output power capacity is very difficult. To solve this problem, a fundamental harmonic of the $TE_{02}$ cylindrical waveguide mode has been applied as the operating mode in our recent design. The interaction cavity radius of the $TE_{02}$ mode is 3.7 mm, which greatly decreases the risk of the bombardment on the tube wall of the electrons; furthermore, the performance of the tube is guaranteed effectively.

The paper is organized as follows. The theoretical analysis of the beam–wave interaction is introduced in Section 2. The third section introduces the designs of the W-band $TE_{02}$ mode gyro-TWT. The performance of the proposed tube is verified by experiment results and the problem of parasitic mode oscillation of the tube is analyzed in Section 4. Finally, the conclusion of the article and the overview of future work are presented in Section 5.

## 2. Theoretical Analysis

In the gyro-TWT, the amplification is achieved by using a cyclotron electron beam to interact with the transmitting microwave, which can be described by the following dispersion equations [22,23]:

$$\omega = k_z v_z + s\Omega \tag{1}$$

$$\omega^2 - k_z^2 c^2 - k_{mn}^2 c^2 = 0 \tag{2}$$

$$k_{mn} = \chi_{mn}/r_w \tag{3}$$

Here, $\omega$ is the operating angle frequency, $k_z$ is the axial wave number, $s$ is the cyclotron harmonic number, $v_z$ is the axial velocity, and $\Omega = eB/(\gamma m_0)$ is the relativistic cyclotron frequency.

The interaction cavity radius can be determined by:

$$r_w = c\chi_{mn}/\omega \tag{4}$$

In the above, $\chi_{mn}$ is the $n$th root of the derivative with respect to $x$ of $J_m(x)$, the $m^{\text{th}}$-order Bessel function of the first kind. The $\chi_{mn}$ of the $TE_{02}$ mode is 7.016, which is roughly two times larger than the $\chi_{mn}$ of the $TE_{01}$ mode. The interaction cavity radius is increased significantly at the same operating frequency. However, the number of competing modes will also increase (shown in Figure 1). Figure 1 shows that the possible competing modes are $TE_{22}$, $TE_{12}$, $TE_{01}$, $TE_{21}$ and $TE_{11}$; meanwhile, they will oscillate at frequencies of 85.5 GHz, 76.2 GHz, 72 GHz, 70.9 GHz, 69.2 GHz and 67 GHz, respectively. Therefore, the method of suppressing these competing modes is critical in the design.

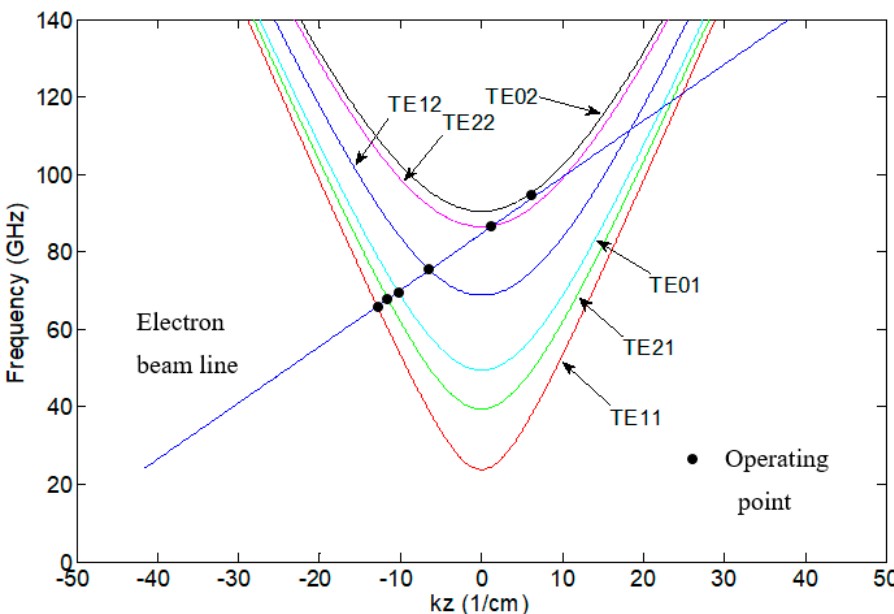

**Figure 1.** The dispersion diagram of the $TE_{02}$ mode and the possible oscillating modes ($U$ = 70 kV, $\alpha$ = 1.2, $B/B_g$ = 1.026, and $r_w$ = 3.7 mm).

According to the observations from the previous experiment for the W-band $TE_{01}$ mode gyro-TWT, in order to suppress these competing modes, a periodic dielectric loaded circuit (shown in Figure 2) has been applied, in which the relative permittivity and the loss tangent of the dielectric, the periodicity, and the dimension of the circuit have been optimized for the highest efficiency.

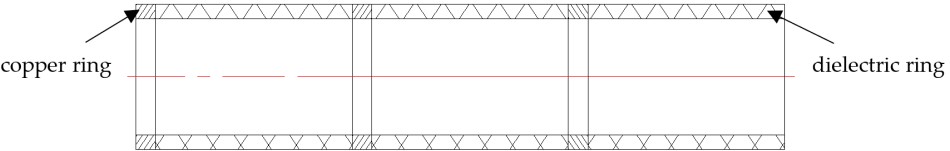

**Figure 2.** The schematic of the periodic dielectric loaded circuit.

Meanwhile, the harmonic coupling coefficient that represents the interaction strength is defined as [22]:

$$H_{sm}(r_g, r_L) = J^2_{s \pm m}(x_{mn} r_g / r_w) J'^2_s(x_{mn} r_L / r_w) \tag{5}$$

Here, $r_L$, $r_g$, and $r_w$ are the Larmor radius, the guiding-center radius of the electrons, and the interaction cavity radius, respectively, with $\pm$ indicating co-rotating ($-$) and counter-rotating ($+$) modes, respectively. Figure 3 shows the dependence of the coupling coefficient on guiding-center radius for the $TE_{02}$ mode and the possible competing modes. In this figure, the coupling coefficient peaks at $r_g/r_w = 0.26$ for the $TE_{02}$ operating mode. Although this is smaller than the coupling coefficient of the $TE_{01}$ mode, the smaller guiding-center radius contributes to the increase in electron flow rate. In addition, the coupling coefficients of the other possible competing modes are very small; the value of $r_g/r_w$ is 0.26, which indicates the oscillations of these modes have less influence on the performance of the tube.

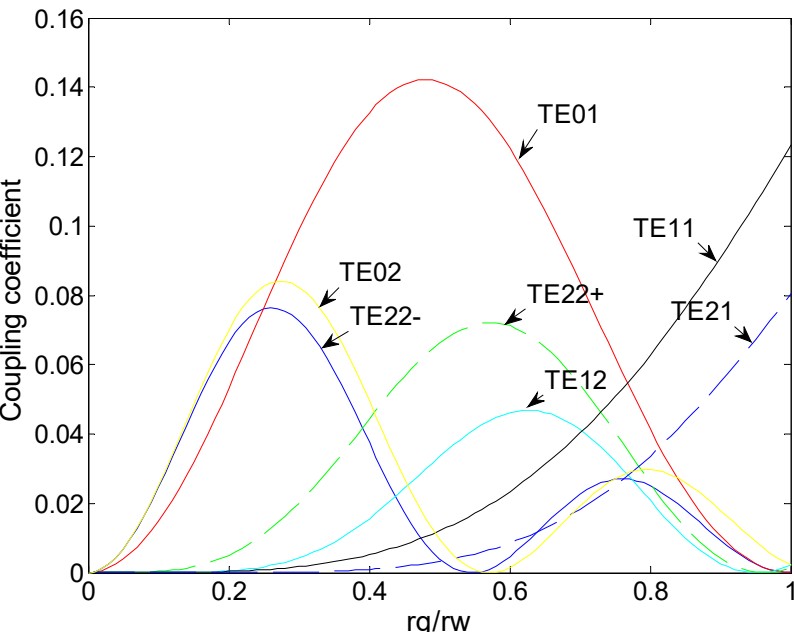

**Figure 3.** The dependence of the coupling coefficient on guiding-center radius for the $TE_{02}$ mode and possible competing modes.

## 3. Designs and Simulation Results

In general, a gyro-TWT includes a magnetic injection gun, an input coupler, an interaction region, a collector, and an output window. The prototype of the W-band $TE_{02}$ mode gyro-TWT is shown in Figure 4. In our design, a triode-type magnetic injection gun with a high voltage of 70 kV and a relatively low beam current of 9 A has been applied. The input coupler consists of a pillbox window and a Y-type mode convertor. The interaction region uses a periodic beryllium oxide ceramic-loaded circuit. The collector is cooled by water. The output window is formed by three pieces of sapphire discs. The operating parameters are listed in Table 1.

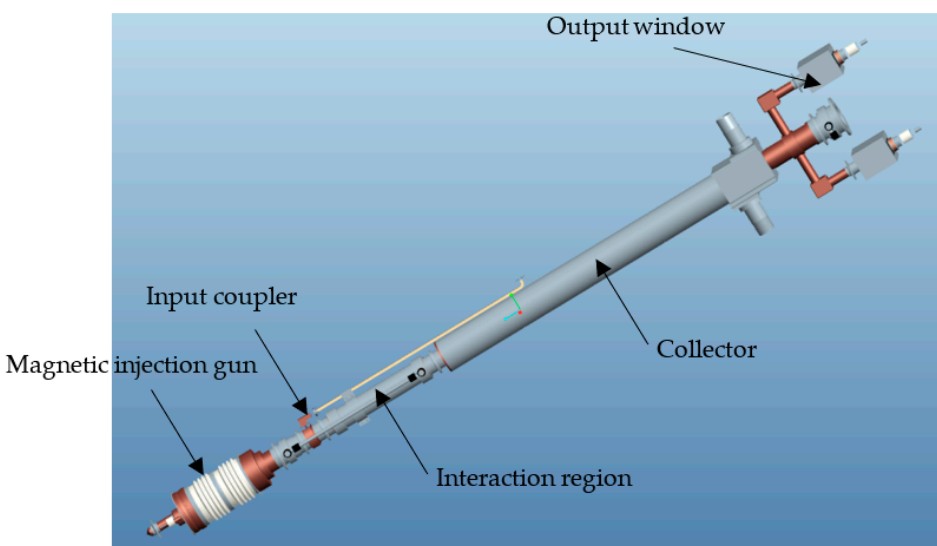

**Figure 4.** Schematic of the W-band $TE_{02}$ mode gyro-TWT.

**Table 1.** Operating parameters of the W-band TE$_{02}$ mode gyro-TWT.

| Parameter | Value |
|---|---|
| Accelerating voltage | 70 kV |
| Electron beam current | 9 A |
| Velocity ratio | 1.2 |
| Operating mode | TE$_{02}$ |
| Output mode | TE$_{01}$ |
| Cyclotron harmonic | 1 |
| Magnetic field | 3.4 T |
| Interaction cavity radius | 3.7 mm |
| Guiding-center radius | 0.96 mm |
| Relative permittivity of dielectric | 12 |
| Loss tangent of dielectric | 0.225 |
| Velocity spread | 5% |

### 3.1. Interaction Region

The interaction region is the most important component of the gyro-TWT, where the energy of the electron beam has been transferred to the RF wave when the synchronous condition is satisfied. According to the analysis in Section 2, the parasitic mode oscillations need to be suppressed. When the higher order mode is applied, the influence of these oscillations on the output power of the operating mode increases significantly. Figure 5 shows the process of the beam–wave interaction at a frequency of 94 GHz. By introducing the periodic dielectric loaded circuit, the parasitic mode oscillations are suppressed. In the smooth waveguide, the power of the input RF wave is amplified from 10 mW to 160 kW with an electron efficiency of 29%.

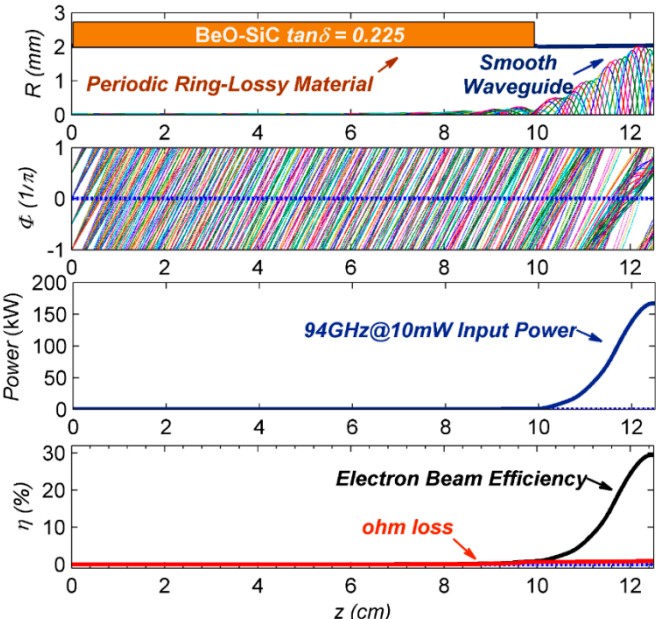

**Figure 5.** Process of the beam–wave interaction in the designed interaction region (*f* = 94 GHz, drive power is 10 mW, velocity spread is 5%).

Figure 6 shows the calculated instantaneous bandwidth of the W-band TE$_{02}$ mode gyro-TWT. The results show that the 120 kW-level bandwidth is at 8.65 GHz, which satisfies the requirements of the application.

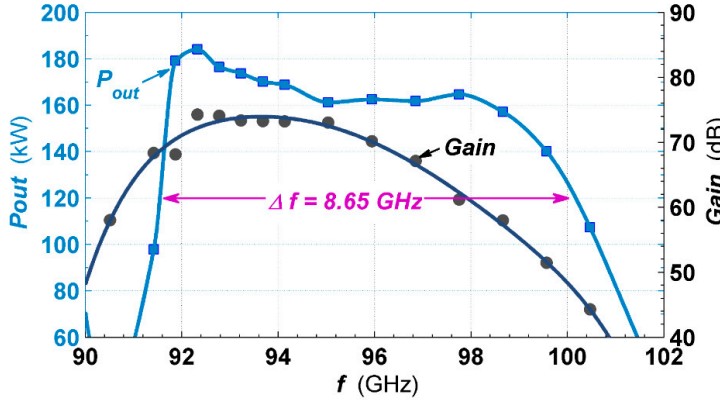

**Figure 6.** Calculated bandwidth of the W-band TE$_{02}$ mode gyro-TWT.

### 3.2. Electron Optical System

The electron optical system consists of: (1) a magnetic injection gun (MIG) for generating the desired rotating electron beam; (2) a collector, which is used to sort the residual electrons after interaction. This particular section deals with the design studies on the electron optical system, which are as follows.

#### 3.2.1. Magnetic Injection Gun

A triode-type MIG consists of an emitter ring, a modulating anode and an accelerating anode. It is preferred for performance verification experiments by adjusting the voltage of the modulating anode to control the electron beam parameters. According to the beam–wave interaction calculation, the velocity ratio, guiding-center radius, and velocity spread are set as the design objectives of the MIG. By optimizing the structure dimensions and the potentials of the electrodes, the desired beam parameters were achieved, and carried out by Opera-3D code (seen in Figure 7 and Table 2).

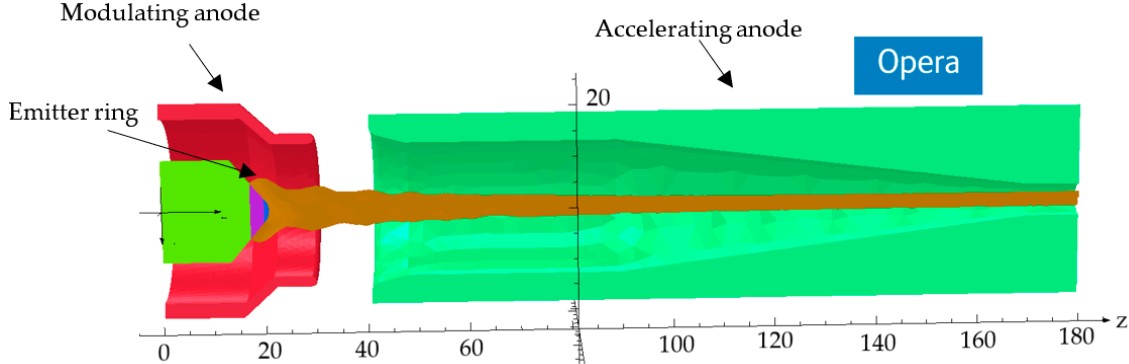

**Figure 7.** Electron trajectories of the optimized MIG.

**Table 2.** Beam properties of the optimized MIG.

| Parameter | Value |
|---|---|
| Accelerating voltage | 70 kV |
| Modulating voltage | 30.7 kV |
| Electron beam current | 9 A |
| Velocity ratio | 1.2 |
| Guiding-center radius | 0.972 mm |
| Velocity spread | 3.49% |

### 3.2.2. Collector

In order to take out the heat generated by the dissipation power of the residual electrons in time, the collector is cooled by water. The optimization of the cooling structure was carried out by ANSYS. Under the parameters of water flow at 7 t/h and dissipation power density at 490 W/cm$^2$ (assuming all the electron energy transforms to heat and the duty ratio is 20%), the results of the temperature distribution are shown in Figure 8. The temperatures of the inner wall and outer surface are 136 °C and 94 °C, respectively, which are far below the outgassing temperature of the copper. According to the calculation, the average power capacity of the designed collector is above 130 kW.

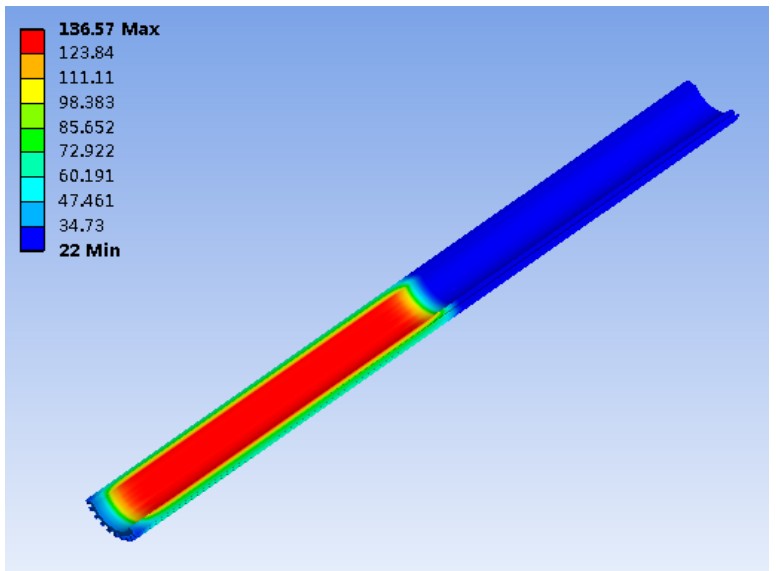

**Figure 8.** Temperature distribution of the optimized collector.

### 3.3. Transmission System

The transmission system includes: (1) an input coupler, which is used to transform the rectangular waveguide TE$_{10}$ mode to the cylindrical waveguide TE$_{02}$ mode; (2) an output window, which is used to maintain a high-vacuum environment in the tube and transmit the cylindrical waveguide TE$_{01}$ mode to the outside of the tube with low reflection. The bandwidth and reflection coefficient are the most important properties of the transmission system.

### 3.3.1. Input Coupler

In our design, the input coupler consists of a pillbox window and a mode converter. The pillbox window is made of CVD diamond, which has a perfect matching performance in a W band. In the mode converter, a Y-type structure has been applied [24–26]. A standard rectangular waveguide has been divided into eight ports to connect with a cylindrical waveguide; these ports are uniformly distributed along the circumference. The simulation results show that the TE$_{02}$ mode can be excited directly from the TE$_{10}$ mode.

Figure 9 shows the calculated VSWR of the input coupler. From 90 GHz to 102 GHz, the VSWRs are between 1.2 and 2.1, which meet the requirements of the broadband power drive. The fluctuation of the VSWR is caused by the reflections of the eight ports.

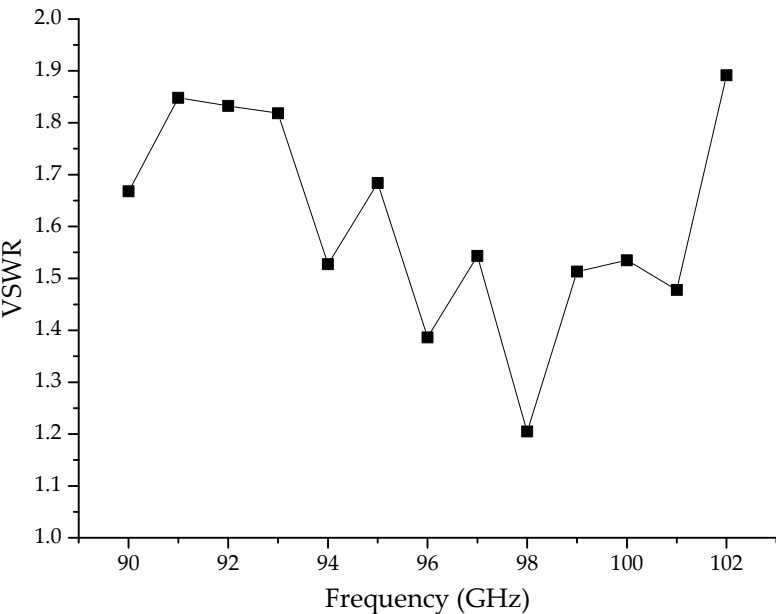

**Figure 9.** Calculated VSWR of the input coupler.

### 3.3.2. Output Window

To achieve broadband operation, a triple-sapphire-disc configuration has been applied on the output window [24]. Based on the view of the resonant window, the thickness of the disc is about $N \bullet \lambda_d / 2$, where $N$ is equal to 1,2,3 . . . , and $\lambda_d$ is the wavelength of sapphire. By optimizing the thicknesses of the discs and the distances between the discs, the lowest reflection coefficient was achieved.

The diameter of the output window is 32 mm. After optimization, the thicknesses of the three discs are 0.4 mm, 0.8 mm, and 0.4 mm, respectively. The distances between the discs are 0.51 mm. The relative dielectric constant and the loss tangent of the sapphire are 9.5 and $2 \times 10^{-4}$, respectively. Figure 10 shows the calculated VSWR of the output window. In 91–100 GHz, the VSWRs are less than 1.1, which indicate that the advantageous property of low reflection is obtained.

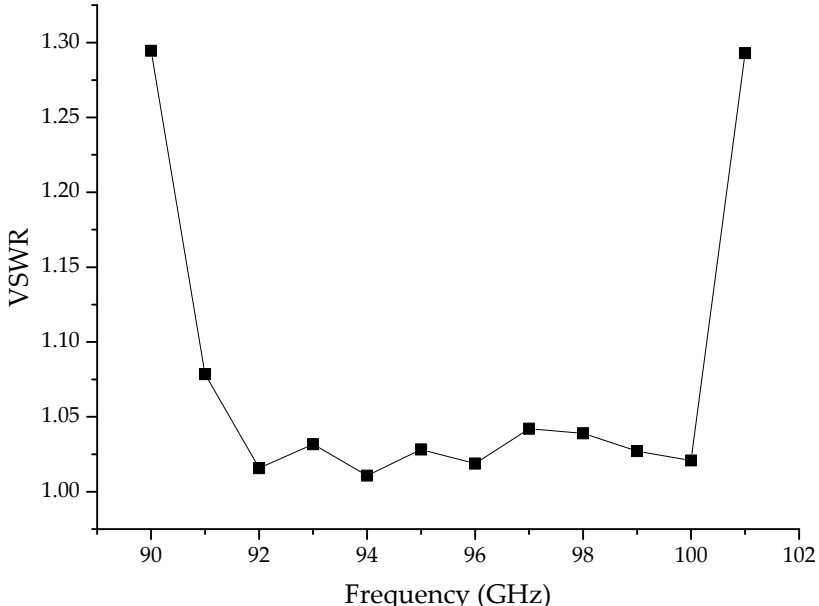

**Figure 10.** Calculated VSWR of the output window.

## 4. Experiment Results and Discussion

According to the design scheme of the W-band TE$_{02}$ mode gyro-TWT, fabrication of the components was carried out. Before assembling the tube, the critical components, such as the input coupler and output window, were tested by the VNA (Vector Network Analyzer), as seen in Figure 11. The cold test results are shown in Figure 12. In 92–100 GHz, the measured VSWRs of the input coupler are less than 2, and the measured VSWRs of the output window are less than 1.1. In consideration of the influences of the machining errors, the above results agree well with the simulation results.

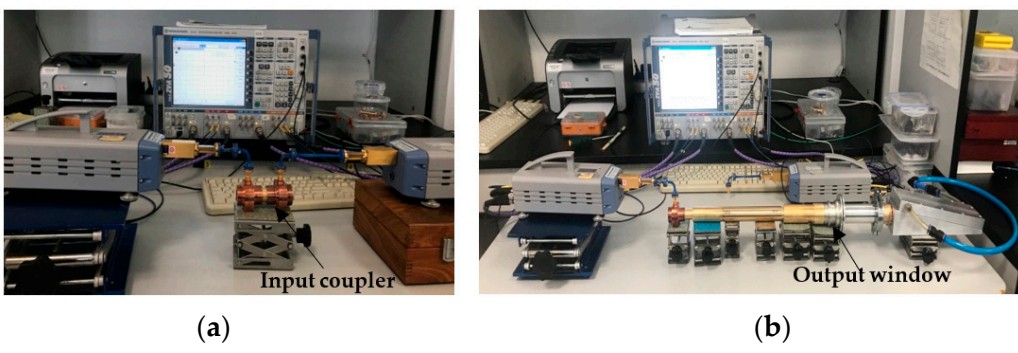

**Figure 11.** Cold test of the critical components of the W-band gyro-TWT by VNA (**a**) Input coupler; (**b**) Output window.

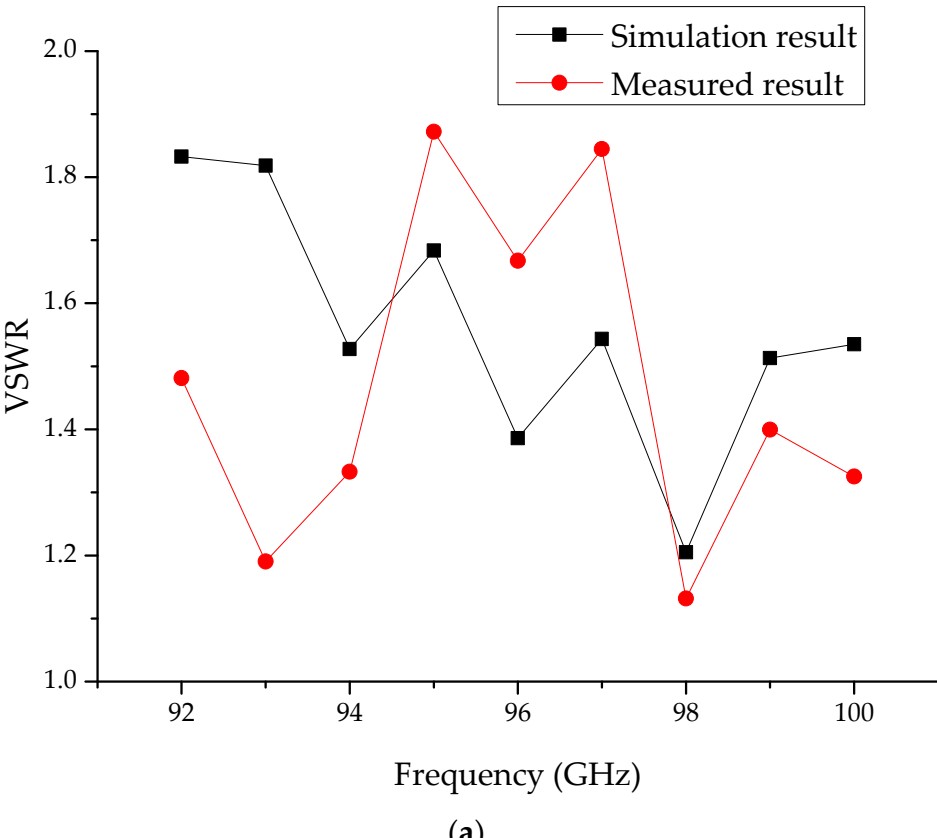

**Figure 12.** *Cont.*

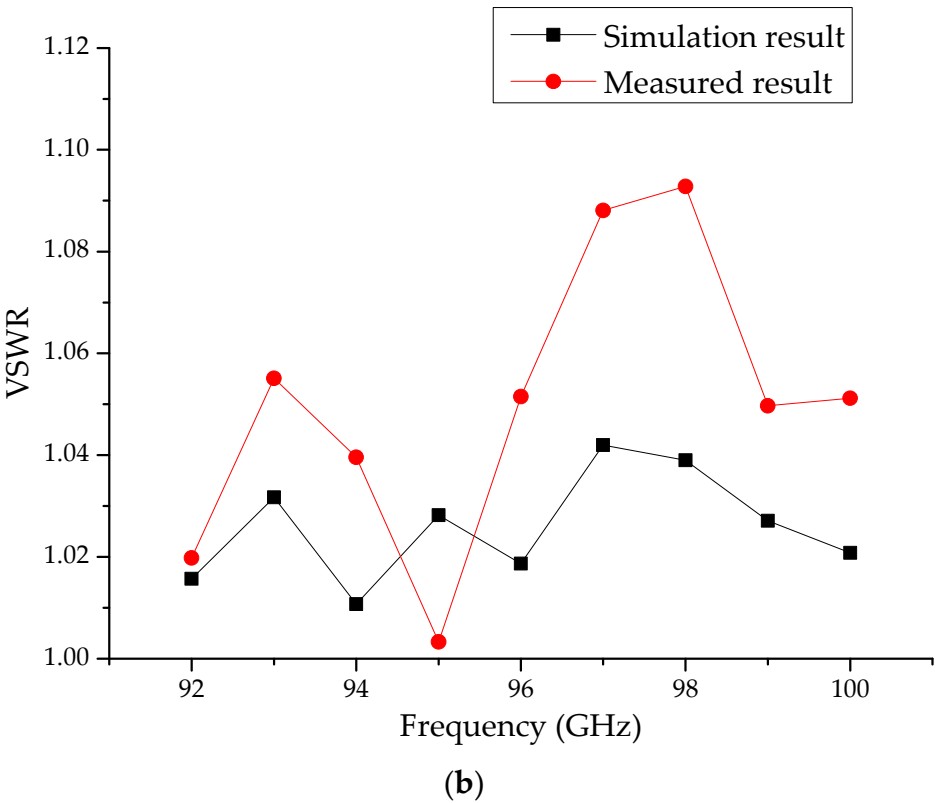

(**b**)

**Figure 12.** Cold test results of the critical components of the W-band gyro-TWT. (**a**) Input coupler; (**b**) Output window.

The prototype of the W-band $TE_{02}$ mode gyro-TWT is shown in Figure 13. The length and maximum diameter of the tube are 1400 mm and 85 mm, respectively.

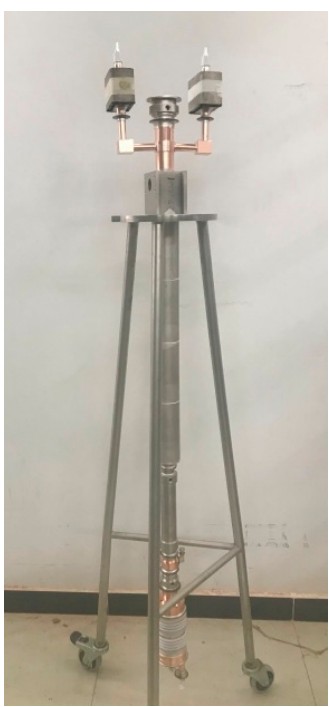

**Figure 13.** Prototype of the W-band $TE_{02}$ mode gyro-TWT.

The hot test system diagram of the tube is shown in Figure 14. A driving signal which is generated by a W-band signal generator is amplified by a W-band solid state amplifier and a W-band TWT. The maximum output power of the W-band TWT is 150 W. The W-band gyro-TWT was installed on a superconducting magnet with a maximum magnetic field intensity of 4 T. The voltage of the W-band gyro-TWT is provided by a high-voltage power supply. The output power of the tube is measured by a water load and a calorimeter. The waveforms of the beam voltage, beam current, driving signal, and output signal are measured by a four-channel oscilloscope. A spectrum analyzer is used to detect the operating mode and parasitic modes.

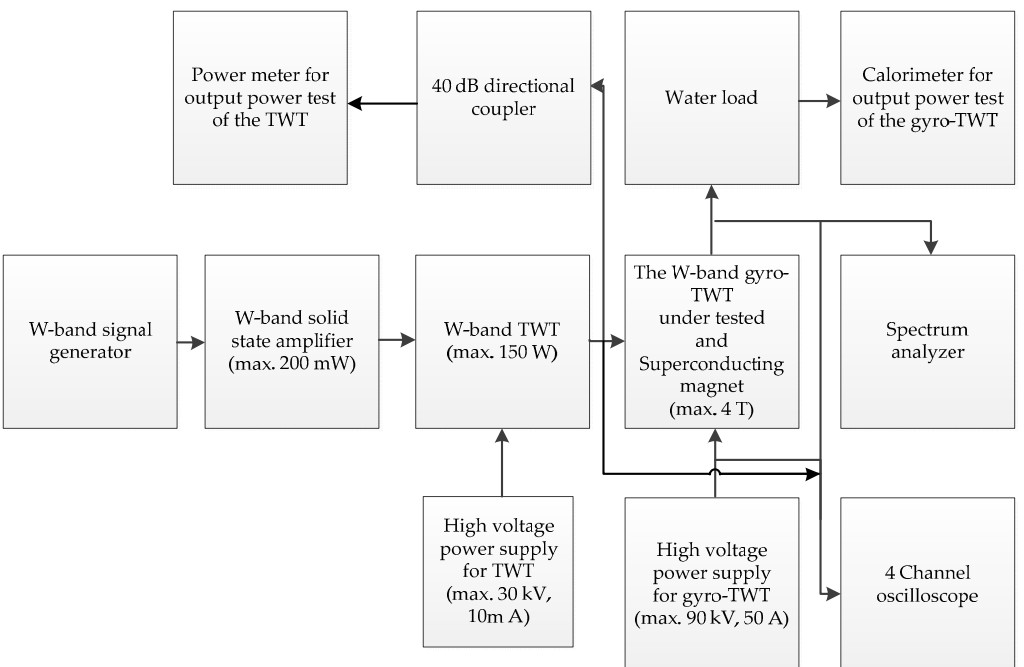

**Figure 14.** Hot test system diagram of the W-band $TE_{02}$ mode gyro-TWT.

First, experimental verification was carried out. In the duty ratio of 1%, the accelerating voltage was 70.2 kV, the electron beam current was 8.5 A, and the maximum magnetic field was 3.46 T. The measured output power and gain are shown in Figure 15. It can be seen that from 91 to 99 GHz, the output power is above 60 kW, and the gain is above 32 dB. The property of broadband power amplification of the W-band $TE_{02}$ mode gyro-TWT was verified.

However, the output power and the gain are much lower than the simulation results, due to the oscillations in the tube. The frequency spectrums of the output signal were detected in the experiment by a spectrum analyzer, as shown in Figure 16. The amplification of the operating mode and the oscillations were observed. According to the dispersion diagram in Figure 1, the oscillations correspond to the absolute instability of the $TE_{02}$ mode at 88 GHz and the backward wave oscillation of the $TE_{22}$ mode at 85 GHz, respectively. These oscillations disturb the movements of the electron beams and further lead to the increase in velocity spread. Therefore, the beam–wave interaction efficiency decreases rapidly. In addition, they have a significant influence on the operating stability of the tube, which restricts the increase in duty ratio. In subsequent improvements, a heavier dielectric loss will be applied in the interaction region to suppress the mode oscillation.

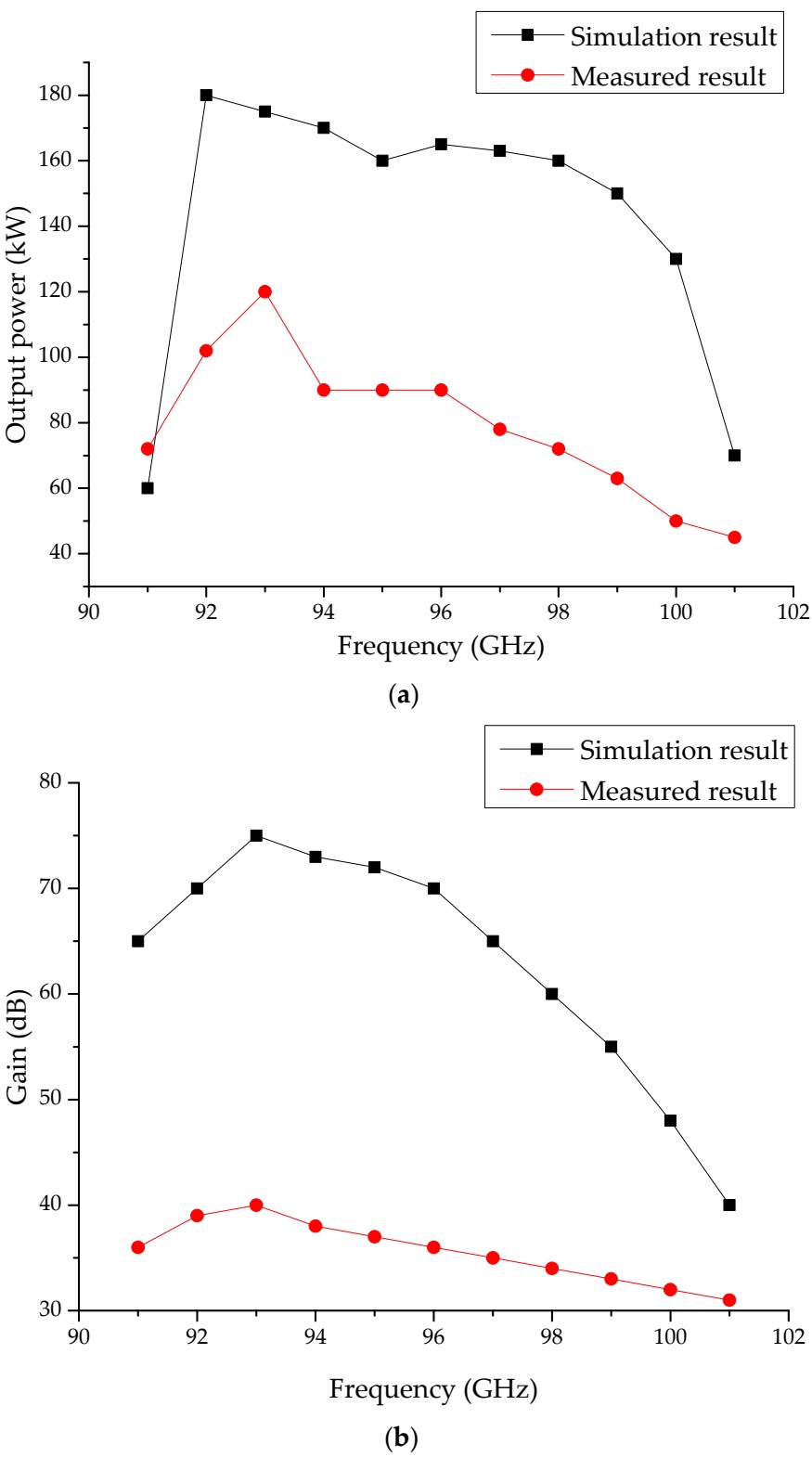

**Figure 15.** Measured output power (**a**) and gain (**b**) of the W-band TE$_{02}$ mode gyro-TWT.

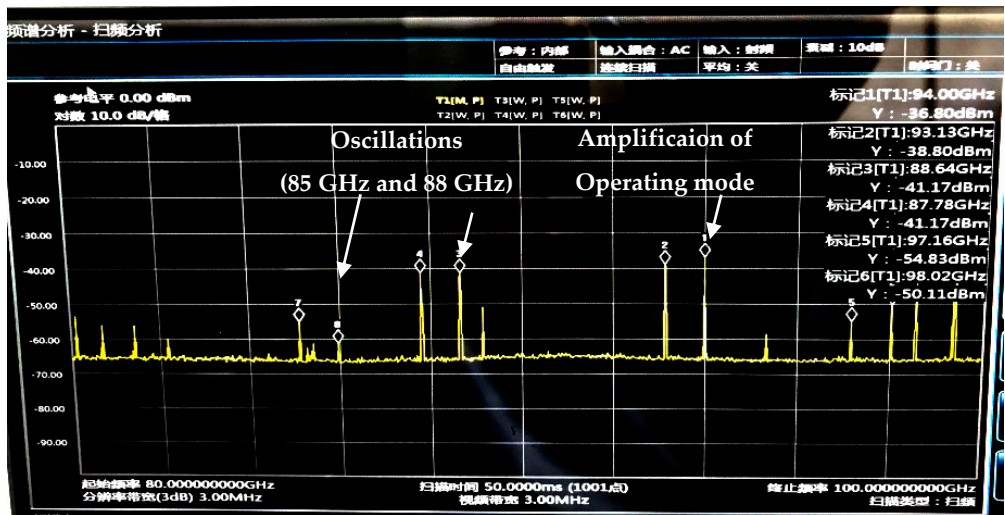

**Figure 16.** Measured frequency spectrums of the W-band TE$_{02}$ mode gyro-TWT.

## 5. Conclusions

In this paper, the design of a W-band TE$_{02}$ mode gyro-TWT is presented. The simulation results indicate that the performances of the tube meet the requirements of the output power greater than 100 kW and the bandwidth greater than 8 GHz. Fabrication of the components was carried out. The cold test results of the transmission system agree well with the simulation results, which verify that the transmission system has the advantageous properties of broadband and loss reflection. The prototype of the W-band TE$_{02}$ mode gyro-TWT was assembled. In the preliminary experiment, a bandwidth of 8 GHz was achieved, and the output powers were above 60 kW at 70.2 kV/8.5 A. The feasibility of using the TE$_{02}$ mode as the operating mode to achieve broadband amplification in the W-band is verified. However, the measured results are far below the simulation's results, which are caused by the oscillations in the tube. The mode oscillations are observed, which causes the beam–wave interaction efficiency to decrease rapidly and the duty ratio to be restricted to 1%. An improvement of applying a heavier dielectric loss to suppress the oscillations is in progress. The reasons for this and the suppression methods are also under study.

**Author Contributions:** Conceptualization, X.Z., C.D. and J.F.; methodology, C.D.; validation, X.Z., C.D., Y.Z.; formal analysis, A.L., S.G., Z.W.; investigation, X.Z., C.D., Z.W. and Z.Z.; resources, J.F.; data curation, X.Z. and Z.W.; writing—original draft preparation, X.Z. and Z.W.; writing—review and editing, X.Z. and J.F.; visualization, Y.Z.; supervision, J.F.; project administration, X.Z.; funding acquisition, X.Z. All authors have read and agreed to the published version of the manuscript.

**Funding:** This research received no external funding.

**Data Availability Statement:** All data included in this study are available upon request by contacting with the corresponding author.

**Conflicts of Interest:** The authors declare no conflict of interest.

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
