# Peer review of "Design and Preliminary Experiment of W-Band Broadband TE02 Mode Gyro-TWT"

_electronics, doi:10.3390/electronics10161950_

Round 1

Reviewer 1 Report

The paper is interesting, however details and precisions are required to properly assess the achievements.

Fig. 1 : not clear/no size/scale indicated

In Table 1, the loss tangent seems quite high, is this loss tangent measured under same field level ? Does it has an impact on performances ?

Fig. 10 caption : I guess this is coupler rather than couple. Why the VSWR is oscillating a lot ? Is there any way to smooth this behavior ?

Fig. 11 : the output window is simulated with the TE01, however the design is done for TE02. It is not very clear for me.

In figure 16, the gain is far lower compared to simulations (75-> 35 dB at 94 GHz). Oscillations are mentionned as a source of problem. Thus, a temporal analysis or a spectrum analysis of the gyro-TWT should be conducted to see how any modulation is affecting (frequency of modulation, index, …) to assess if it can really explain the big gain reduction.

Also, output power is quite high (> 100 kW). With a 35 dB gain, the device should be pumped with 100 W at 94 GHz. How was this conducted in terms of power calibrations ? In general, how are measured the high powers here ?

Fig. 17. The spectrum shown is a snapshot photo taken with a phone I guess. A graph with good shape would be better for a MDPI paper in my opinion.

Fig. 16. Why the output power is really decreasing after 93 GHz ? Any modal change ?

Generally speaking, more discussion on instrumental procedure, power/VNA calibrations, all of this operated over pulsed mode, would be necessary, because this is not straightforward in theses power levels/frequency bands. This is important for work cross-comparisons within the field. Overall the paper is interesting.

Reviewer 2 Report

Many thanks for the important results presented in this manuscript. Despite the very interesting results presented, the reviewer detected a major inconsistency with regards to the claimed gain (70 dB) and input power (10 mW) versus the input power (150 W) used in the experiment. Additinally, the authors might consider to structure their manuscript as the authors are starting with a finding from experiments, then switching to fundamental gyrotron theory, the switching to the design, and, finally switching back to experiments. As to the reviewer the most important finding is the remaining oscillations in the tube. This should be discussed in more details and concluded in more details in the summary. What makes the authors confident to solve it by using different dielectrics? The Authors might consider to enlarge some pictures, particular Fig. 13. The authors might consider to include a space between numbers and units. 

Author Response

Thank you very much for your suggestions.

First, 150W is the output power of the TWT, which has been attenuated by the transmission lines between the TWT and the gyro-TWT. By cold test, the total insert loss of the transmission lines is 12dB. Therefore, the practical value of the drive power is less than 10W. However, the oscillations have a significantly influence on the parameters of the electron beams, which leads to the beam-wave interaction efficiency is far below the simulation result.

Second, to solve the problem of the W-band TE01 mode gyro-TWT, the design of the W-band TE02 mode gyro-TWT has been carried out in our institute. The oscillations have not been detected in the simulations, but it has been observed in the first experiment, which is beyond our expectation. Therefore, we hope to suppress the oscillations by increasing the loss of the dielectric. Meanwhile, we are studying on the reasons and the suppression methods of the oscillations. We will report the research results as soon as possible.

Finally, according to your suggestion, the Fig. 13 has been adjusted.

Reviewer 3 Report

The described work is sound and only minor modifications are suggested.

Despite that the developed W-band Gyro-TWT was developed for a specific high-resolution imaging radar, the authors are kindly requested to provide a better description of the state of the art in gyrotron traveling wave tubes in  terms of achieved power and working frequency.

The practical performance of the output window is an important detail of the gyro-TWT especially at high power level. The authors are kindly requested to provide data on the performance of the sapphire discs used for output window.

Author Response

Thank you very much for the suggetions.

According to your suggestion, the description of the state of the art in gyro-TWT has been adjusted.

The relative dielectric constant and the loss tangent of the sapphire are 9.5 and 2×10-4 respectively.

Reviewer 4 Report

Thanks for working on Gyrotron traveling wave tube. The topic is interesting however the paper is poorly written. In the abstract even there is a lot of language mistakes, please rectify and correct the structuring. Please review the statement " It can be applied to imaging radar to improve the resolution and operating range". How does it improve as the pulse doesn't have a very short duration?

Keywords should be in alphabetical order. 

there must be no figures in the introduction. The only explanation is enough. Please review the whole introduction again.

Give references to the basic equations.

Figure 4: Review the axis font and coupling coefficient must be Coupling coefficient.

How the Pout is calculated?

figure 8, 9: fonts must e visible if printed.

Figure 10, 11: It is highly unprofessional to directly put snapshot from HFSS. review a good papers and see how the plots are presented. This is an important plot and must be of high quality.

The VSWR is not stable at W-band? Of course it is less than 2.

figure 13: same issue as above.

simulation and measurement need to be correlated to see the difference. It must be in the same plot.

Similarly measured and simulated Pout must be plotted in same plot. Dont plot it in excel. Use professional software including matlab or origin.

Rewrite the conclusion by removing mistakes and information from simulation and measurement comparison.

Author Response

Thank you very much for your suggestions.

Round 2

Reviewer 4 Report

Thanks for revising the manuscript. It is improved considerably however the grammatical and typo mistakes are still there which need to be addressed very carefully.

Line 79: please check ref [22] did not give a proper insight of the equations. Modify the reference and be careful.

Figure 3: Axis is still small and there is an extra t on the y-axis so eliminate it.

Point 6: How the Pout is calculated?
Response 6: Pout is calculated by the numerical code based on the linear theory of the gyrotron which is compiled by us. The reference is as follows.
E. Borie and B Jodicke. Comments of the linear theory of the gyrotron. IEEE Transactions on plasma science, vol. 16. no.2, 1988. 

It would be interesting if you give the GitHub link or any related means.

Thanks

Author Response

(The authors gave the same response as above.)
